# Human Amniotic Membrane Mesenchymal Stem Cell-Synthesized PGE_2_ Exerts an Immunomodulatory Effect on Neutrophil Extracellular Trap in a PAD-4-Dependent Pathway through EP2 and EP4

**DOI:** 10.3390/cells11182831

**Published:** 2022-09-10

**Authors:** Gibrán Alejandro Estúa-Acosta, Beatriz Buentello-Volante, Fátima Sofía Magaña-Guerrero, José Eduardo-Aguayo Flores, Oscar Vivanco-Rojas, Ilse Castro-Salas, Karla Zarco-Ávila, Mariana A. García-Mejía, Yonathan Garfias

**Affiliations:** 1Cell and Tissue Biology, Research Unit, Institute of Ophthalmology Conde de Valenciana, Mexico City 06800, Mexico; 2Department of Biochemistry, Faculty of Medicine, Universidad Nacional Autónoma de México, Mexico City 04510, Mexico

**Keywords:** NET, immunomodulation, prostaglandin E_2_, amniotic membrane, mesenchymal stem cells, EP2, EP4, PAD4

## Abstract

Human amniotic membrane mesenchymal stem cells (hAM-MSC) secrete a myriad of components with immunosuppressive activities. In the present research, we aimed to describe the effect of prostaglandin E_2_ (PGE_2_) secreted by hAM-MSCs on neutrophil extracellular trap (NET) release and to characterize the role of its receptors (EP2/EP4) in PAD-4 and NFκB activity in neutrophils. Human peripheral blood neutrophils were ionomycin-stimulated in the presence of hAM-MSC conditioned medium (CM) treated or not with the selective PGE_2_ inhibitor MF-63, PGE_2_, EP2/EP4 agonists, and the selective PAD-4 inhibitor GSK-484. NET release, PAD-4, and NFκB activation were analyzed. Ionomycin induced NET release, which was inhibited in the presence of hAM-MSC-CM, while CM from hAM-MSCs treated with MF-63 prevented NET release inhibition. PGE_2_ and EP2/EP4 agonists, and GSK-484 inhibited NET release. EP2/EP4 agonists and GSK-484 inhibited H3-citrullination but did not affect PAD-4 protein expression. Finally, PGE_2_ and EP2/EP4 agonists and GSK-484 increased NFκB phosphorylation. Taken together, these results suggest that hAM-MSC exert their immunomodulatory activities through PGE_2,_ inhibiting NET release in a PAD-4-dependent pathway. This research proposes a new mechanism by which hAM-MSC exert their activities when modulating the innate immune response and inhibiting NET release.

## 1. Introduction

The amniotic membrane (AM) is the innermost layer of the placenta that is in contact with the fetus. The AM is an avascular and translucent tissue composed of a simple epithelium and a fibrous matrix embedded with mesenchymal stem cells (MSC). The secretome of the MSC isolated from human AM (hAM-MSC) has been widely studied, demonstrating its capacity to modulate the immune system [1,2]. Conditioned medium from hAM-MSC (hAM-MSC-CM) generates an immunosuppressor microenvironment suppressing B lymphocyte activity [3], directing macrophages toward an anti-inflammatory phenotype [4] and decreasing neutrophil tissue infiltration [5]; these results indicate that hAM-MSC-CM possesses bioactive molecules that exert their anti-inflammatory action, such as TSG-6 [6], sHLA-G [7], and prostaglandins (PG) [8], among many others.

PGE_2_ is an eicosanoid derived from arachidonic acid that has paradoxical (pro- and anti-inflammatory) effects in different biological systems. PGE_2_ promotes an anti-inflammatory neutrophil phenotype, induces interleukin (IL)-10 production from macrophages, directs macrophages into an anti-inflammatory phenotype, inhibits IL-23 and IL-12 production by human monocytes, downregulates MMP-9 and iNOS in macrophages, suppresses inflammasome formation, regulates M1 macrophages, increases regulatory T cells, and generates regulatory IL-10-dependent dendritic cells [9].

Neutrophils are the most abundant leukocytes in the peripheral blood and constitute the first line of defense against microorganisms, exerting their functions such as phagocytosis, degranulation of their preformed components, synthetization of radical oxygen species (ROS), and release of extracellular traps (NET). In addition to its antimicrobial functions [10], NET release is strongly associated with inflammatory-mediated diseases [11]. NET are extracellular DNA fibers decorated with granular proteins such as neutrophil elastase (NE) and myeloperoxidase (MPO) [12]. Citrullinated histones are also found as NET components [13]. There are two main pathways associated with NET release, NADPH (NOX)-dependent and NOX-independent pathways, depending on the stimulus [14]. Thus, phorbol 12-myristate 13-acetate (PMA), which directly activates protein kinase-C (PKC), which in turn phosphorylates NOX, is the main stimulus related to NOX-dependent pathway NET release, while ionophores such as ionomycin activate the NOX-independent pathway, stimulating peptidyl arginine deiminase (PAD)-4, a calcium-dependent enzyme that increases H3-citrullination; promoting chromatin decondensation [15]; and actively participating in NET release. Both pathways are, in a certain way, mutually inhibitors or complementary. Although there is increasing research on NET release events, important features are still missing.

PGE_2_ inhibits NET release in a PGE_2_-PGE_2_-receptor (EP)-2- and/or PGE2-EP4-dependent manner in PMA-induced NET release models. Both EP2 and EP4 augment intracellular levels of cAMP and increase PKA activity, inhibiting NET release [16,17]. This background describes the effects of EP2 and EP4 on NET release inhibition in PMA-induced (NOX-dependent) NET release; however, the participation of EP2 and EP4 in ionomycin-induced (NOX-independent) NET release is unknown. The present study shows the effect of hAM-MSC-CM on ionomycin-induced NET release. This study identifies PGE2 as a soluble factor secreted by hAM-MSC involved in NET release inhibition and describes the crosstalk between EP2 and EP4 and PAD-4 activity.

## 2. Materials and Methods

### 2.1. Reagents

Polymorphoprep was purchased from Alere Technologies AS (Jena, Germany); ethylenediaminetetraacetic acid (EDTA) was obtained from Promega (Madison, WI, USA); calcium chloride fused granular was purchased from Merck (Darmstadt, Germany); vectashield-4′,6-diamidino-2-phenylindole (DAPI) was obtained from Vector Laboratories (San Diego, CA, USA); and PGE_2_, Butaprost (BUT), CAY10598 (CAY), GSK-484 (GSK) and 2-(9-chloro-1H-phenanthro (9,10-d)imidazol-2-yl)-1,3-benzenedicarbonitrile (MF-63) were obtained from Cayman Chemical Company (Ann Arbor, MI, USA). Anti-histone H3Cit (citrulline R2+R8+R17 (ab5103)), anti-histone H3 (ab1791), anti-PAD4 (ab50247), anti-neutrophil elastase (ab21595), anti-SSEA-4 (ab16287), anti-Oct-4 (ab18976), anti-tubulin (ab6047), anti-NFκB (ab16502) and anti-pNFκB (ab86299), antibodies were purchased from Abcam (Cambridge, MA USA). Alexa Fluor 488- and 594-conjugated goat anti-rabbit antibodies were obtained from Life Technologies (Eugene, OR, USA). Syto 14-Green was purchased from Thermo Fisher (Waltman, MA USA). PerCP-Cy 5.5-conjugated mouse anti-human CD15, PE-conjugated mouse anti-human CD11b FITC-conjugated anti-human CD29, FITC-conjugated anti-human CD90, PE-conjugated anti-human CD105, and PE-Cy7-conjugated anti-human CD45 antibodies were obtained from BD Bioscience (San Jose, CA, USA). HRP-conjugated goat anti-rabbit secondary antibodies were obtained from Jackson-Immuno-Research Laboratories (West Grove, PA, USA), and a PGE2 Parameter Assay Kit was purchased from R&D Systems (Minneapolis, MI, USA). Other reagents were purchased from Sigma-Aldrich (Saint Louis, MI, USA), unless otherwise stated.

### 2.2. Neutrophil Isolation and Stimulation

Neutrophils were isolated from healthy human blood donors using polymorphoprep according to the manufacturer’s instructions. Briefly, equal quantities of blood and polymorphoprep were centrifuged at 320× *g* for 35 min at 20 °C. The cells were carefully obtained and washed with ice-cold Hanks’ balanced salt solution. The trypan blue exclusion method was used to determine neutrophil viability, obtaining > 95% cell viability. Neutrophils were seeded on poly-*L*-lysine charged glass coverslips in 96-well plates in Hankߣs solution. The plates were incubated for 20 min at 37 °C to allow adhesion of the cells, after which neutrophils were stimulated with ionomycin from *Streptomyces conglobatus* (1.25 µM) and calcium chloride (1.5 mM) for 1 h at 37 °C in a 5% CO_2_ atmosphere.

### 2.3. Treatments of Neutrophils

Neutrophils were pretreated with 10 nM PGE_2_, an EP2 receptor agonist (Butaprost), or an EP4 receptor agonist (CAY10598) for 15 min before stimulation with ionomycin-calcium, and GSK-484 was added 30 min prior to stimulation in some experiments.

### 2.4. Isolation and Characterization of Human Amniotic Mesenchymal Stem Cells (hAM-MSC)

The present research was approved by the IRB with number CEI-2021/06/04. Isolation and characterization of hAM-MSC were performed according to our previous reports [6,18].

### 2.5. Preparation of Conditioned Medium (CM)

Conditioned medium (CM) was prepared according to our previous report, with slight modifications [6]. hAM-MSC were treated with or without MF-63 (10 μM), and the supernatants were harvested at different culture times; this was considered the conditioned medium (CM).

### 2.6. PGE2 Quantification

PGE2 quantification in CM from hAM-MSC was performed using the Prostaglandin E2 Parameter Assay Kit according to the manufacturer’s instructions. Briefly, the samples were diluted according to the assay procedure. Primary antibody solution was added to each well of the microplate, and the samples were incubated for 1 h at RT. PGE_2_ conjugate was added to each well, covered, and incubated for 2 h at RT on a shaker. Next, the cells were washed four times, the substrate solution was added, and the cells were incubated for 30 min at RT. Finally, stop solution was added, and the optical density of each well was determined. Absorbance was measured at 450 nm with a microplate reader. A standard curve was created, and we analyzed the obtained data by Prism 5 Graph Pad software ver. 8.0.2 (La Jolla, CA, USA).

### 2.7. Immunofluorescence Assays

Neutrophils (4 × 10^4^) were allowed to settle for 15–20 min onto glass coverslips and placed in the wells of 24-well plates at 37 °C in 5% CO_2_. The cells were treated with ionomycin-calcium and incubated for 1 h at 37 °C.

To analyze NET release, stimulated neutrophils were fixed with 4% paraformaldehyde for 10 min at RT. Next, the cells were washed with PBS 1X three times and then blocked with blocking solution (PBS 1X 0.01% Triton X-100 and 5% BSA) for 1–2 h at RT with gentle shaking. Afterwards, the samples were incubated with the purified primary antibodies anti-neutrophil elastase (NE) (1:100) or anti-citrullinated histone (H3Cit) (1:300) overnight at 4 °C with gentle shaking. On the next day, the coverslips were washed twice with the wash buffer (PBS 1X and 0.05% Tween 20 [PBS-T]); subsequently, the samples were incubated for 2 h in darkness with Alexa Fluor 488 goat anti-rabbit antibody (1:800) at RT with gentle shaking. Extracellular DNA was visualized with propidium iodide (PI, (0.025 mg/mL)) incubating for 5 min at RT in darkness. Tthe coverslips were rinsed twice with PBS and mounted with PBS 1X with glycerol (1:1). The images were acquired with an Apotome II microscope, recorded using AxioVision 2.0 software Carl Zeiss(Jena, Thüringen) and analyzed using ImageJ/Fiji software (Stapleton, Nueva York, USA).

### 2.8. NET Quantitation

To quantify NET, images were acquired with an ApoTome II microscope using ZEN 3.4 (blue edition) software from Carl Zeiss(Jena, Thüringen c). We used machine learning for the detection and quantification of NET. A classification model was created with the deep learning module Intellesis Trainable Segmentation Carl Zeiss(Jena, Thüringen). Four images were randomly selected for detection training, the training module was set to multispectral mode, and three classes were created for classification: background, nuclei, and NET. To normalize the NET area measured in each field, we chose only fields with 80 nuclei stained with DAPI. Nuclear detection was automatically performed with the ZEN cell nuclear counting module. We set up 50-feature deep learning and conditional random field postprocessing for image segmentation. Starting from the final detection model, we set up the image analysis module, and three randomly selected images from three independent trials were analyzed. Finally, the NET release area was obtained.

### 2.9. Western Blotting Analysis

To determine the expression of H3Cit, PAD4 and p-NFκB, human neutrophils (1.9 × 10^6^) under the aforementioned conditions were lysed, and Western blotting was performed. Briefly, cell lysates were separated by 4–20% SDS-PAGE, and proteins were blotted onto nitrocellulose membranes. The membranes were blocked for 1 h 30 min at RT with 5% BSA or 5% milk in TBS-T (TBS 1X and 0.1% Tween 20). Subsequently, the membranes were incubated with a dilution of primary antibodies at 4 °C. The membranes were rinsed with TBS-T and incubated with goat anti-rabbit secondary Ab conjugated to HRP for 2 h at RT, and finally, the membranes were washed three times with TBS-T. HRP activity was detected using a chemiluminescence substrate (Super Signal West Pico or West Femto, Thermo-Scientific, Rockford, IL, USA).

### 2.10. Statistics

Data were assessed by one-way ANOVA followed by Bonferroni correction test and, in some cases, Mann-Whitney tests. The data are expressed as the mean ± standard error of the mean (SE), and values of *p* < 0.05 were considered statistically significant. All statistics were performed using Prism 5 Graph Pad software ver 8.0.2 (La Jolla CA, USA).

## 3. Results

### 3.1. hAMSC-CM Decreased NET Release

In the Supplementary images (Appendix A), human neutrophils and human amniotic membrane stem cell (hAM-MSC) characterization are shown; moreover, we have previously described the stemless of these cells [6,18]. We previously reported that CM obtained from hAM-MSC inhibited NET release in a murine model [6]; in this context, we attempted to identify whether similar results were obtained in NET release in a human neutrophil model using a calcium ionophore such as ionomycin as an inducer. Neutrophils were stimulated with ionomycin for 1 h prior to staining with specific antibodies against neutrophil elastase (NE) and PI. The results showed that although human neutrophils released NET in a basal manner, ionomycin significantly enhanced NET release; in contrast, when neutrophils were incubated with ionomycin in the presence of hAM-MSC-CM of 12 h, NET release was significantly inhibited (*p* < 0.05) (Figure 1).

### 3.2. CM from MF-63-Treated hAM-MSC Prevented NET Release Inhibition

MF-63 is an inhibitor of microsomal prostaglandin synthase-1 (mPGES-1); thus, its action is to inhibit PGE_2_ synthesis. hAM-MSC were incubated with MF-63 at different time points, and CM was obtained to determine the effect of PGE_2_ absence on NET release. hAM-MSC-CM obtained from 3 and 6 h with and without MF-63 did not affect NET release. Interestingly, MF-63-nontreated hAM-MSC-CM obtained at 9 and 12 h significantly inhibited NET from ionomycin-treated neutrophils compared with MF-63-treated hAM-MSC-CM, which prevented NET release inhibition, suggesting that PGE_2_ is responsible for inhibiting NET release (Figure 2). Moreover, to quantify PGE_2_ in MF-63-treated and untreated hAM-MSC-CM, ELISA was performed, and we observed that hAM-MSC-CM constitutively secreted PGE_2_, while MF-63 treatment reduced PGE_2_ secretion (Figure 2).

### 3.3. Exogenous PGE_2_ Inhibited NET Release in Ionomycin-Stimulated Neutrophils

Ionomycin-stimulated neutrophils were cultured in the presence of different concentrations of PGE_2_ (10, 100, and 1000 nM). The NET released were visualized with extracellular DNA staining with Syto 14 green dye. The results showed that PGE_2_ was able to inhibit NET release at 10 nM PGE_2_ (Figure 3A,B). To corroborate the inhibition of NET release by PGE_2_, neutrophil elastase (NE) staining, a component of NET, was performed. The 10 nM concentration of PGE_2_ was able to reduce NET-associated components such as NE (Figure 3C).

### 3.4. NET Are Reduced with PGE_2_, EP2, and EP4 PGE_2_ Receptor Agonists and GSK-484

Ionomycin treatment significantly induced NET release from neutrophils. As expected, pretreatment with PGE_2_ significantly reduced NET from ionomycin-treated neutrophils, and similar results were obtained when ionomycin-treated neutrophils were pretreated with EP2 and EP4 PGE_2_ receptor agonists and GSK-484 a selective inhibitor of PAD-4 (Figure 4).

### 3.5. EP2 and EP4 Affected PAD-4 Activity and NFκB Phosphorylation, While PAD-4 Protein Expression Was Unchanged

Next, we sought to determine whether the EP2 and EP4 PGE2 receptors reduced citrullination of H3. Neutrophils were pretreated with Butaprost (EP2 agonist) or CAY-10598 (EP4 agonist) for 15 min and then incubated with ionomycin for 1 h. Western blot analysis showed that ionomycin increased H3 citrullination in comparison to non stimulated conditions, while EP2 and EP4 PGE_2_ receptor agonists were able to significantly reduce H3 citrullination. Moreover, neutrophils preincubated with the selective PAD-4 activity inhibitor GSK-484 inhibited H3 citrullination. Unexpectedly, PGE_2_ was unable to reduce H3-citrullination in pretreated ionomycin-stimulated neutrophils. Moreover, hAM-MSC-CM was able to downregulate PAD-4 activity, inhibiting citrullination of H3 (Appendix A). On the other hand, ionomycin significantly inhibited NFκB phosphorylation, while PGE_2_ and EP2 and EP4 PGE_2_ agonist receptors as well as GSK-484 prevented ionomycin-NFκB phosphorylation inhibition in neutrophils. Interestingly, none of the aforementioned conditions affected PAD-4 protein expression (Figure 5).

## 4. Discussion

In the present study, we showed that hAM-MSC-CM is able to inhibit neutrophil activity in terms of inhibiting NET release. Additionally, we were able to identify that hAM-MSC-synthetized PGE_2_ is, in part, the eicosanoid responsible for reducing NET release. Moreover, the PGE_2_ main activity in this context was driven by EP2 and EP4 receptors. Finally, we showed that EP2, EP4, and GSK inhibited PAD-4 activity and that PGE_2_, EP2, EP4, and GSK upregulated NFκB phosphorylation, which inhibited NET release. In a previous study, we described that hAM-MSC synthetized and secreted TSG-6, an anti-inflammatory molecule that is able to reduce the membrane mitochondrial potential, which is related to NET release in an in vitro model using murine neutrophils [6]. Moreover, in another study, we demonstrated that hAM-MSC-CM has antifibrotic and anti-inflammatory activity in an alkali corneal burn in an in vivo model [18]. This background prompted us to study the effect of hAM-MSC-CM on NET release in a more detailed manner. First, NET release was reduced in the presence of CM from hAM-MSC, while CM from hAM-MSC cultured in the presence of MF-63 prevented NET release inhibition. Cyclooxygenase-2 (COX-2) generates the common PG precursor PGH_2,_ while microsomal prostaglandin E2-synthase-1 (mPGES-1) produces PGE_2_; thus, employing a selective mPGES-1 inhibitor such as MF-63, we selectively inhibited PGE_2_ synthesis without affecting other PHG_2_-derived products. This result indicated that PGE_2_ is, in part, responsible for NET inhibition release. PGE_2_ is an eicosanoid synthesized from arachidonic acid with ambivalent (pro- and anti-inflammatory) functions. PGE_2_ is able to decrease T-cell proliferation [8], induce an anti-inflammatory macrophage phenotype [19], and inhibit inflammasome activation through EP2 and EP4 [20], among many other anti-inflammatory effects. Although MF-63 inhibits PGE_2_ synthesis and promotes anti-inflammatory effects by upregulating IL-1a and downregulating IL-6 [21], a clearer system is needed to confirm the direct effect of PGE_2_ on NET release inhibition. To corroborate that the anti-inflammatory effect exerted by hAM-MSC-CM was mediated by PGE_2_ secretion, neutrophils were directly exposed to exogenous PGE_2_. Interestingly, we observed that exogenous PGE_2_ was able to significantly reduce ionomycin-induced NET release. Previously, it has been described that PGE_2_ is able to inhibit PMA-induced NET release in an EP2-(EP4)-cAMP dependent mechanism [16,17]. In contrast to the aforementioned studies, in our research, we induced NETosis using the calcium ionophore ionomycin; therefore, this stimulus activates a NOX-independent NETosis pathway, while PMA activates a NOX-dependent NETosis pathway. These two pathways are in a certain manner opposite among them, as has been previously described [15]. In the former, activation of the Ca^2+^-dependent PAD-4 enzyme and a (leukotoxic) hypercitrullination (LTH) process are observed, while in the PMA-dependent NETosis pathway, the LTH process is inhibited and PAD-4 is not affected, but PKC activity is enhanced. Thus, our experiments with ionomycin allowed us to identify the mechanism driven by PAD-4-dependent H3-citrullination. In this context, it has been demonstrated that autoantibodies against citrullinated proteins are essential in autoimmune-mediated diseases such as rheumatoid arthritis; therefore, using ionomycin as a NET inducer helped us to identify whether PGE_2_ is able to inhibit the PAD-4 citrullination process.

PGE_2_ exerts its biological activity mainly through four recognized receptors (EP1-EP4) [22]. From those receptors, EP2 and EP4 have been shown to possess immunosuppressive effects, enhancing the cAMP-PKA signaling pathway [23]. In this context, we performed assays employing high affinity EP2 and EP4 agonists, such as Butaprost and CAY-10598, respectively. Butaprost and CAY-10598 were able to significantly reduce H3 citrullination, suggesting that there is crosstalk between EP2 and EP4 receptor signaling and PAD-4 activity. Interestingly, hAM-MSC-CM was also able to inhibit PAD-4 activity (Appendix A). However, a crosstalk between EP2 and PE4 receptors signaling, and PAD-2 activity cannot be ruled out [24]. Whether this reduction in citrullination is directly caused by EP2 and EP4 receptors is still a matter for further studies. Unexpectedly, PGE_2_ alone was not able to significantly reduce H3 citrullination; in this context, independent EP2-EP4 PGE_2_ activity cannot be ruled out [25]. Moreover, a functional selectivity of synthetic and natural ligands of EP4 has been described given by their ligand-receptor affinity constants [26], which could explain the inequal effect observed among PGE_2_, and EP2 and EP4 receptors in H3 citrullination. Both EP2 and EP4 are G protein-coupled receptors, and their signaling pathway leads to an increase in intracellular cAMP through Gα_s_ protein that increases the gene expression of NFκB complex components [27]; therefore, the NFκB signaling pathway is activated. In terms of stimulus, in our research, ionomycin significantly reduced NFκB phosphorylation; this result is in accordance with that previously reported by Brignall R et al., who described that ionomycin induced NFAT activation rather than NFκB activation [28]. Interestingly, PGE_2_, EP2/EP4 agonists, and GSK-484 prevented ionomycin NFκB activation inhibition, confirming that PGE_2_ activates NFκB; in this context, it has been described that NFκB plays a role in the repression of PAD4 transcription [29]. Although we did not analyze PAD4 mRNA, western blotting revealed that PAD4 was not affected in any condition, which suggests that NFκB activity could act at different protein levels, for example, inhibiting PAD4 enzymatic activity instead of PAD4 expression.

## 5. Conclusions

Human amniotic membrane mesenchymal stem cells synthetize and secrete PGE_2_, which is able to modulate the innate immune response, inhibiting NET release through the downregulation of PAD-4. In the present research, we proposed crosstalk between EP2/EP4 and PAD-4 activity, suggesting a new mechanism by which hAM-MSC exert their activities in modulating the innate immune response.

## Figures and Tables

**Figure 1 cells-11-02831-f001:**
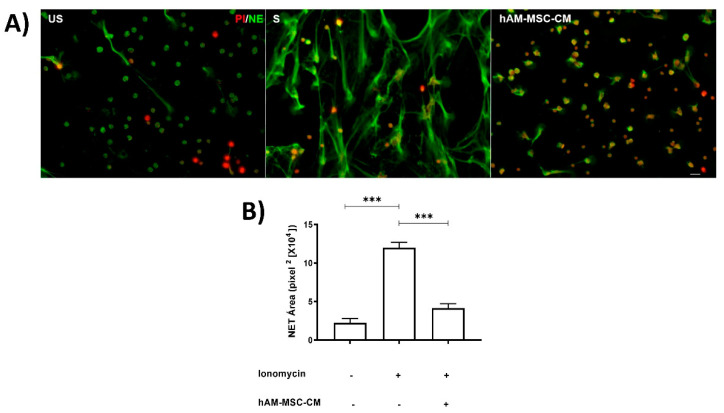
**hAM-MSC-CM****decreased NET release.** Human neutrophils were stimulated with ionomycin (1.25 µM) for 1 h to induce NET release and incubated with hAM-MSC-CM. Fluorescence micrographs of unstimulated neutrophils (US, left panel), ionomycin-stimulated neutrophils (S, middle panel), and ionomycin-stimulated neutrophils incubated with hAM-MSC-CM of 12 h (right panel). Scale bar, 20 µm. These are representative images from three independent assays (**A**). Graphical representation of NET release under the aforementioned conditions (**B**). NET release was quantified as described in the Materials and Methods section. Bars represent the mean ± SE. *** *p* < 0.001. US, unstimulated; S, stimulated; hAM-MSC-CM, human amniotic membrane conditioned medium.

**Figure 2 cells-11-02831-f002:**
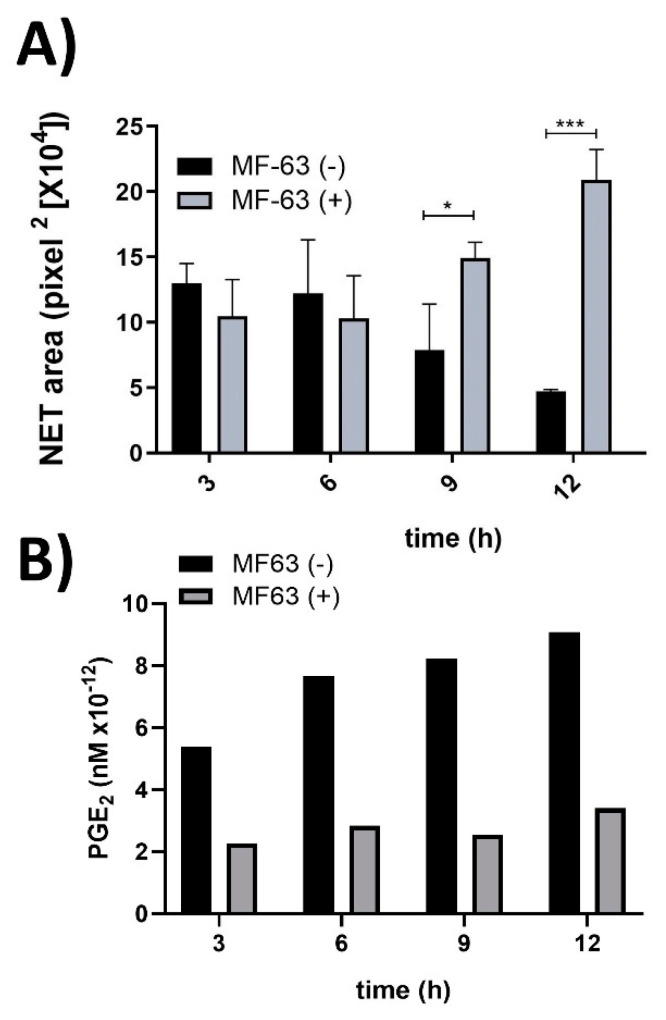
**PGE_2_ secreted from hAM-MSC decreased NET release.** NET from ionomycin (1.25 µM)-stimulated neutrophils were inhibited in the presence of hAM-MSC-CM without MF-63 (10 µM) treatment at 9 and 12 h, while this inhibition was prevented with hAM-MSC-CM obtained after MF-63 treatment. The results are expressed as the mean of 3 independent assays performed in triplicate. Bars represent the mean ± SE, * *p* < 0.5; *** *p* < 0.001. CM (conditioned medium) was collected at the indicated time points from hAM-MSC incubated with MF63 (gray bars) or without MF-63 (black bars) (**A**). Levels of PGE_2_ analyzed in hAM-MSC-CM with and without MF-63 treatment. hAM-MSC were cultured in the presence (white bars) and absence (filled bars) of MF-63 for the indicated times. The PGE_2_ concentration was measured by ELISA (**B**).

**Figure 3 cells-11-02831-f003:**
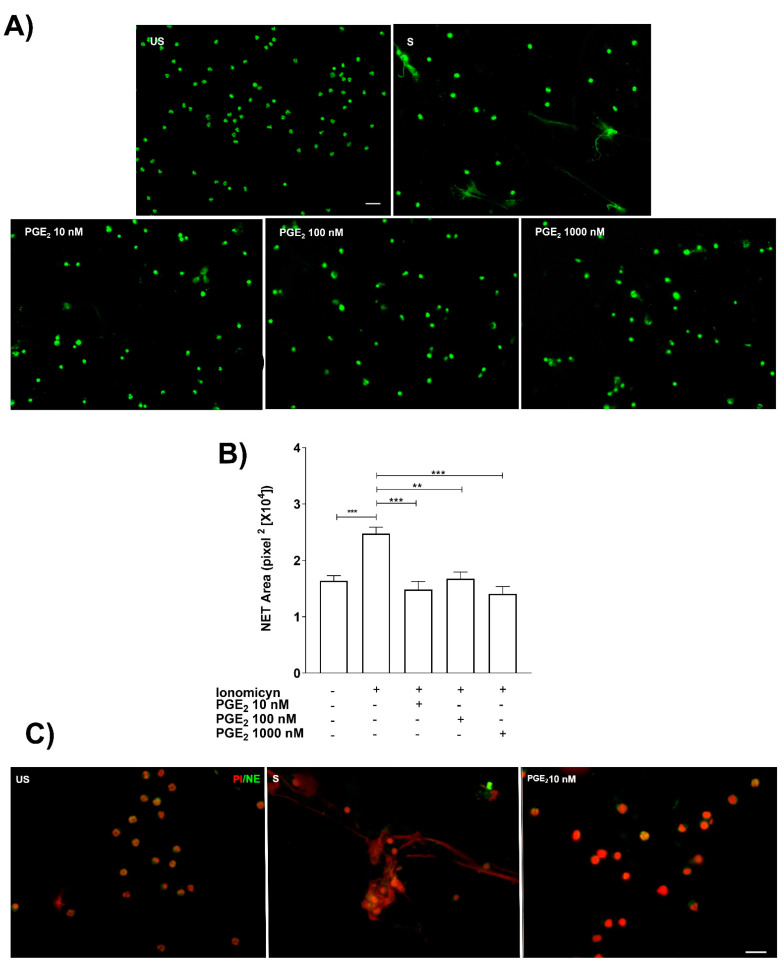
**Exogenous PGE_2_ inhibits NET release.** Representative microscopy images from neutrophils pretreated (15 min before) with different PGE_2_ doses (10–1000 nM) and stained with Syto 14-green dye. Scale bar 100 μm (**A**). PGE_2_ significantly inhibited NET release from ionomycin (1.25 µM)-treated neutrophils. Bars represent the mean ± SE from five different independent assays. ** *p* < 0.01, *** *p* < 0.001 (**B**). Representative microscopy images with anti-neutrophil elastase (NE) corroborated that PGE_2_ (10 nM) inhibited NET release. These are representative images from three independent assays. Scale bar 20 µm. PI stands for propidium iodide (**C**).

**Figure 4 cells-11-02831-f004:**
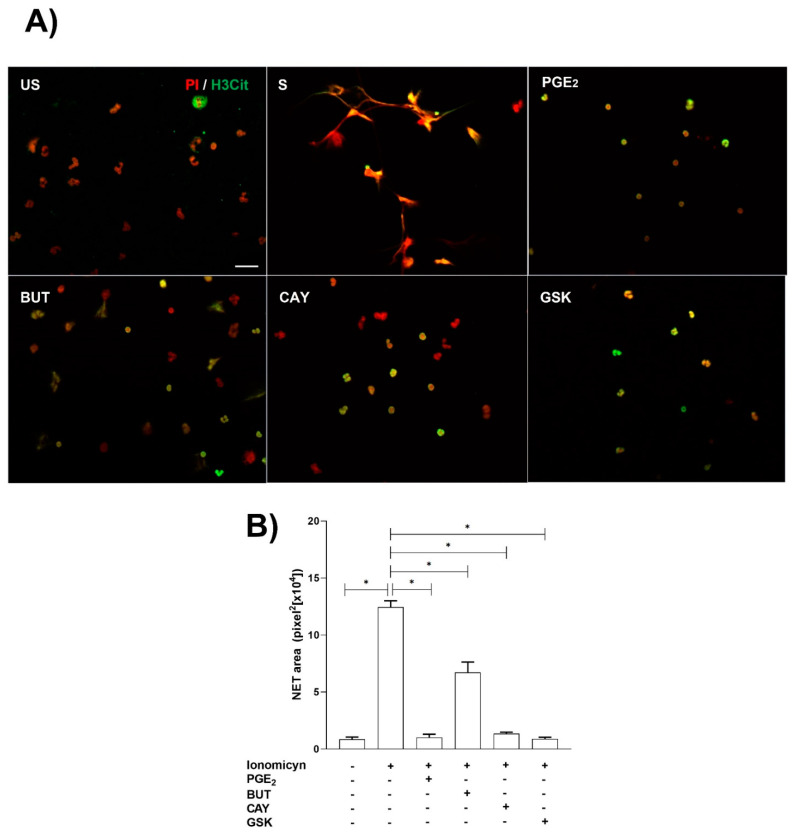
**NET release is reduced by PGE_2_, EP2, and EP4 PGE_2_ receptor agonists and GSK-484.** Neutrophils were preincubated with PGE_2_ (10 nM), Butaprost (BUT) (10 nM), CAY-10598 (CAY) (10 nM), and GSK-484 (10 µM) 15 min before ionomycin stimulation (S) and stained with H3-citrullinated (H3Cit) and propidium iodide (PI). The NET area was quantified in the obtained images. Scale bar, 20 µm (**A**). Ionomycin induced NET release, while PGE_2_ reduce NET released. However, CAY and GSK-484 significantly (*p* < 0.05) inhibited NET release (**B**). Bars represent the mean ± SE. * *p* < 0.05.

**Figure 5 cells-11-02831-f005:**
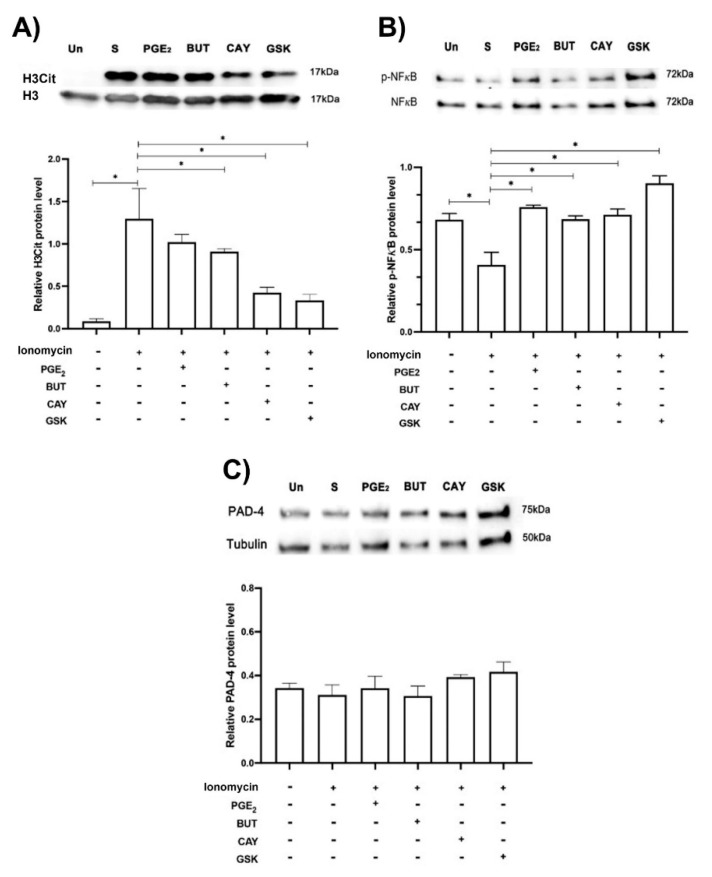
**EP2 and EP4 agonists affected H3 citrullination and NF****κB phosphorylation in ionomycin-treated neutrophils.** Blotting of H3 Cit, PAD-4, and NFκB phosphorylated (p-NFκB) of neutrophils in normal condition (unstimulated condition, Un) and treated with ionomycin (1.25 µM, S) and plus PGE_2_ (10 nM), EP2 (10 nM, BUT, butaprost) and EP4 (10 nM, CAY, CAY-10598) PGE2 agonists, and the PAD-4 selective inhibitor GSK (10 µM, GSK-484). Representative image of blotting and density analysis of H3 Cit (**A**), NFκB phosphorylated (**B**), and PAD-4 (**C**). H3Cit was augmented in the presence of ionomycin (S, stimulated) in comparison with the unstimulated condition (Un), while BUT and CAY (PGE_2_ agonists), and GSK (PAD-4 selective inhibitor) significantly reduced H3Cit. NFκB phosphorylation was significantly reduced with ionomycin, while PGE_2_, BUT CAY, and GSK prevented ionomycin-NFκB phosphorylation inhibition in neutrophils. PAD-4 protein expression was not modified under any of the aforementioned conditions. Representative blots from 5 independent experiments. Bars represent the mean ± SE (lower panels). * *p* < 0.05.

## Data Availability

Not applicable.

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
