# Peer review of "Human Amniotic Membrane Mesenchymal Stem Cell-Synthesized PGE_2_ Exerts an Immunomodulatory Effect on Neutrophil Extracellular Trap in a PAD-4-Dependent Pathway through EP2 and EP4"

_cells, 2022, doi:10.3390/cells11182831_

Round 1
Reviewer 1 Report
My comments are as follow:
1. Have the authors tried to perform staining for RBD-binding B cells in fresh blood (or freshly isolated PBMCs) and compare it to the frequencies of SARS-Cov-2-Spike specific B cell subsets detected when the samples are freezed/thawed?
2. At bottom of the table 1, there are absolute number of T cell and B cell subsets. Does it mean flow cytometry was performed on whole blood samples before freezing down PBMCs?
3. Why the levels of NET area differ a lot between each figure, but the standard errors within each figure are very narrow? Was that due to donor cells variation? If so, do the authors perform experiments in each figure using cells from the same donors (although in the figure legend it stated that the results are from 3-5 independent assays)? Are the results consistent when using different donors?
4. Have the authors quantified the levels of PGE2 in the hAMSC-CM in figure 1 and figure 2? How does it compare to the degrees of NET reduction by directly adding the exogenous PGE2 in figure 3?
5. Was the NET area measured in each field normalized to cell number based on second staining such as nucleus staining?
6. In figure 4, it doesn’t seem like the EP2 agonist (Butaprost) is able to inhibit NET release?
7. Line 283, please clarify if PGE2 and EP2 and EP4 PGE2-receptor agonists and GSK-484 “reverted” or “prevent/reduced” ionomycin-NFkB phosphorylation inhibition? The current experiment design was to incubate neutrophils with these agonists then stimulate the cells with ionomycin, but no experiments to stimulate neutrophils before co-culture with PGE2-receptor agonists.
8. Figure 5B, the difference of phosphorylated NFkB seems to be minimal among all conditions, including unstimulated and stimulated. I am not convinced for the significance of the changes based on the blot images shown here. Have the authors tried other methods to more accurately quantify NFkB phosphorylation? Also, have the authors tried to measured these parameters (H3 citrullination, NFkB phosphorylation…) in conditions used in previous figures where neutrophils were cultured with hAMSC-CM?
9. Line 302-303, data in figure 5 didn’t show convincing evidence that authors claimed here (“EP2, EP4 and GSK induced a PAD-4 activity inhibition and PGE2, 302 EP2, EP4 and GSK up-regulated NFkB phosphorylation”). Please clarify how the authors came to this conclusion. Especially the authors also stated in the main text that PAD-4 protein expression was not altered by any of the experimental condition.
Reviewer 2 Report
The paper proposes a a new mechanism by which hAM-MSC influences the innate immune response possibly by crosstalk between EP2/EP4 and PAD-4 activity. Although the paper does not lack scientific efforts, the significance of the research does not come across. Furthermore, the sentences in the introduction and abstract are too long and so is the title.
There are some minor edits that I would like to suggest:
• By the other hand, iono- 281 … should be on the other hand
• ionomycin (S, stimulated) in comparison with the unstimulated condition (S), 290
Are they both supposed to be (S)
• driven thorough EP2 and EP4 receptors. 301
“Through”
Author Response
Answers to the Reviewer 2
We wish to thank the reviewer for taking the time and effort necessary to provide the pertinent arguments to improve the quality of the manuscript.
All the issues asked by the reviewer were answered, giving a full explanation here, and the manuscript was modified as needed. All modifications kindly suggested by the reviewer to the original article are now highlighted in yellow. We put into your consideration such changes.
Reviewer 2
The paper proposes a new mechanism by which hAM-MSC influences the innate immune response possibly by crosstalk between EP2/EP4 and PAD-4 activity. Although the paper does not lack scientific efforts, the significance of the research does not come across. Furthermore, the sentences in the introduction and abstract are too long and so is the title.
R= Thank you for your comments. Regarding the sentences in the introduction and abstract that are too long we made and effort to write them shorter without any success, thus to make it comprehensive to the readers we prefer not to change them, including the title which although is too long we consider is self-explanatory.
There are some minor edits that I would like to suggest:
By the other hand, iono- 281 … should be on the other hand
R= Thank you for your observation, we have changed the text as suggested.
ionomycin (S, stimulated) in comparison with the unstimulated condition (S), 290
Are they both supposed to be (S)
R= Thank you for your observation. We amended this mistake.
driven thorough EP2 and EP4 receptors. 301
“Through”
R= Thank you for your observation. We have changed the text as suggested.
Reviewer 3 Report
I have a few minor comments on the structure of the paper. I think the manuscript is promising and with the below recommendations can be suitable for publication.
1. The authors use different abbreviations for human Amniotic Mesenchymal Stem Cells (hAM-MSC and hAMSC) - the name format should be as uniform as possible.
2. How many placenta donors were included in the study? What was the age of the donors?
3. Phenotypic characterization of hAM-MSC: Which markers were used for the cytometric analysis of these cells? Please enter some phenotype.
4. Figures: The X-axis of the chart shows the specific categories being compared. Authors should name each group/bar.
5. Figure 2 was included in the manuscript. However, the authors do not refer to it.
6. Ethical statement number is missing. Authors should include more detail. The name of the approving body and the approval number/ID should be included.
Author Response
Answers to the Reviewer 3
We wish to thank the reviewer for taking the time and effort necessary to provide the pertinent arguments to improve the quality of the manuscript.
All the issues asked by the reviewer were answered, giving a full explanation here, and the manuscript was modified as needed. All modifications kindly suggested by the reviewer to the original article are now highlighted in yellow. We put into your consideration such changes.
I have a few minor comments on the structure of the paper. I think the manuscript is promising and with the below recommendations can be suitable for publication.
The authors use different abbreviations for human Amniotic Mesenchymal Stem Cells (hAM-MSC and hAMSC) - the name format should be as uniform as possible.
R= In the new version of the manuscript, we have used the name format uniformly hAM-MSC through all the text.
How many placenta donors were included in the study? What was the age of the donors?
R= We thank this observation. We performed this research using 2 placentas from two different donors (26 and 32 years old). We obtained the AM tissue from the Amniotic Tissue Bank from the Institute of Ophthalmology Conde de Valenciana Foundation.
Phenotypic characterization of hAM-MSC: Which markers were used for the cytometric analysis of these cells? Please enter some phenotype.
R= We have used the following markers: presence of SSEA-4, Oct-4, CD90, CD29 and CD105 and absence of the panleucocytic CD45 marker, as shown in supplementary figure S2.
Figures: The X-axis of the chart shows the specific categories being compared. Authors should name each group/bar.
R= We thank this commentary, however we consider that due the number of conditions evaluated in each assay, to name each group/bar could be difficult to understand all the graphics and figures. Thus, we consider that is clearer to explain each experimental condition using the specific categories in the X-axis.
Figure 2 was included in the manuscript. However, the authors do not refer to it.
R= Thank you for your observation. In the new version of the manuscript, we have referred to Figure 2 in the main text.
Ethical statement number is missing. Authors should include more detail. The name of the approving body and the approval number/ID should be included.
R= In the new version of the manuscript, we have detailed the requested information regarding ethical approval.
Round 2
Reviewer 1 Report
Data in the current manuscript cannot sufficiently support the claim of hAM-MSCs’ immunomodulatory activities through PGE2 being PAD-4-dependent.
This claim solely came from figure 5, where the PGE2 receptor agonists show some inhibition of H3 citrullination. However, changes in PAD-4 protein levels were not observed. The authors explained that the enzyme activity could be affected although the protein levels were unaltered. While it is a possibility, it would be an overclaim to take it as conclusion without validation. The authors should also rule out the difference in enzyme activity or protein levels of PAD-2, which is another PAD enriched in neutrophils.
Author Response
R= We agree with this reviewer, the enzyme activity and/or protein level of PAD-2 cannot be ruled out in the present research and a statement has been included in Discussion section, page 13 line 349-351.
Sincerely,
Yonathan Garfias
The corresponding author.